# Full-Scale O₃/Micro-Nano Bubbles System Based Advanced Oxidation as Alternative Tertiary Treatment in WWTP Effluents

**Laura Ponce-Robles** [1,*] **, Aránzazu Pagán-Muñoz** [2] **, Andrés Jesús Lara-Guillén** [2] **,**
**Beatriz Masdemont-Hernández** [3] **, Teresa Munuera-Pérez** [3] **, Pedro Antonio Nortes-Tortosa** [1]
**and Juan José Alarcón-Cabañero** [1]

[1] Group of Irrigation, Centro de Edafología y Biología Aplicada del Segura (CEBAS-CSIC), 30100 Murcia, Spain
[2] Technology Centre for Energy and Environment (CETENMA), 30353 Murcia, Spain
[3] SISTEMA AZUD, S.A., 30820 Alcantarilla, Spain
*   Correspondence: lponce@cebas.csic.es

**Abstract:** Wastewater treatment plant effluents can be an important source of contamination in agricultural reuse practices, as pharmaceuticals are poorly degraded by conventional treatments and can enter crops, thereby becoming a toxicological risk. Therefore, advanced tertiary treatments are required. Ozone (O₃) is a promising alternative due to its capacity to degrade pharmaceutical compounds, together with its disinfecting power. However, mass transfer from the gas to the liquid phase can be a limiting step. A novel alternative for increased ozone efficiency is the combination of micro-nano bubbles (MNBs). However, this is still a fairly unknown method, and there are also many uncertainties regarding their implementation in large-scale systems. In this work, a combined O₃/MNBs full-scale system was installed in a WWTP to evaluate the removal efficiency of 12 pharmaceuticals, including COVID-19-related compounds. The results clearly showed that the use of MNBs had a significantly positive contribution to the effects of ozone, reducing energy costs with respect to conventional O₃ processes. Workflow and ozone production were key factors for optimizing the system, with the highest efficiencies achieved at 2000 L/h and 15.9 gO₃/h, resulting in high agronomic water quality effluents. A first estimation of the transformation products generated was described, jointly with the energy costs required.

**Keywords:** alternative tertiary treatment; COVID-19; micro-nano bubbles; ozonation; pharmaceuticals

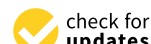



## 1. Introduction

High levels of pharmaceutical manufacturing, and therefore their widespread use, can contribute to their discharge in concentrations high enough to cause adverse effects in aquatic environments, posing a new global challenge [1]. Specifically, from 2010 to 2022, an increase of more than 35% was detected in the consumption and manufacture of pharmaceuticals, and this is expected to increase by 3–6% in 2025 [2]. This increase is attributed to continuously changing demand, which has been impacted in recent years by critical events, including COVID-19 [3]. The main environmental problem is that most of the pharmaceuticals used both at home and in hospitals are excreted from the body either unchanged or as active metabolites at high concentrations, resulting in their continuous release into wastewater collection systems (mainly through wastewater treatment plants, WWTP) [4,5]. However, conventional WWTPs are not effective in the complete removal of these types of compounds (highly variable removal rates, from 0 to 100%). This has given rise to detection of compounds from different therapeutic groups (antibiotics, analgesics, anti-inflammatories, antiretrovirals, etc.) in aquatic environments in recent years, in both developed and developing countries, with values greater than 100 μg/L observed in some cases [6,7]. These concentrations are usually associated with consumption patterns. As an example, the model developed by Kuroda et al. (2021) showed an increase in the

concentration of chloroquine in domestic wastewater due to its consumption during the COVID-19 pandemic, with predicted values of 857 ng/L [8], while various commonly used pharmaceuticals (diclofenac, naproxen, or ketoprofen) were the most detected in WWTP effluents [9]. Although the information on the effects of many of these compounds on the environment and public health remains scarce, and no strict legal regulations on the release of pharmaceuticals into water bodies have been implemented on a global scale, most of them are considered priority substances, and included in European watch lists ((EU) 2022/1307, (EU) 2020/1161, etc.) [10,11].

Furthermore, the problem becomes aggravated when, in order to deal with water stress, the intention is to valorize WWTP effluents through agricultural reuse practices to promote the use of alternative water resources according to the circular economy action plans [12,13]. However, the reality is that the pharmaceutical compounds contained in these effluents could enter the food chain, which may result in health problems due to their potential accumulation risk (chronic effects) [14,15]. In light of this, the application of recently developed tertiary treatments to produce better water quality is currently attracting the attention of researchers and water managers as an opportunity for its reuse for suitable purposes. The challenge is to remove pharmaceutical compounds and pathogens that could have negative effects on human health, animal health, or the environment. In addition, the emergence of new, increasingly restrictive reuse policies (such as the new (EU) 2020/741 on minimum requirements for water reuse, to be implemented at the European level from June 2023) makes investing money in appropriate treatments a short-term necessity, as well as the development of efficient treatments to meet these legal requirements with assumable energy, costs, or other resources [16].

Ozone-based processes are an attractive option that could be implemented as a tertiary treatment in WWTP as ozone, as a result of its strong oxidizing power (2.07 V), is reported to be an effective disinfectant, and an agent capable of degrading pharmaceuticals using two mechanisms: (a) direct electrophilic attack by molecular ozone; and (b) indirect attack by hydroxyl radicals ($\bullet$OH), produced through the ozone decomposition process. It is this path that makes ozonation to be considered part of the classification of advanced oxidation processes (AOPs) [17]. The main advantage of ozone over other AOPs, such as Fenton or photo-Fenton processes, is the fact that the addition of complex chemical reagents (i.e., iron salts or chelating agents) are not needed to carry out the reaction [18].

However, the ozone mass transfer process from the gas to the liquid phase can be a limiting step, resulting in a low-efficiency process. Therefore, ozone coupling with other technologies ($O_3/H_2O_2$, $O_3/UV$, $O_3/ultrasound$, $O_3/electrolysis-ozonation$, or catalytic ozonation) has been studied in recent years [19–22]. In fact, this process seems to be the bottleneck that is hampering the definitive implementation of this technique, given the large investments needed in ozone generation systems.

One innovative technique for enhancing mass transfer and increasing ozone solubility in water is the combination of $O_3$ with micro-nano bubbles (MNBs) [23]. Their tiny size (with diameters ranging from tens of nanometers to several tens of micrometers) and specific properties have captured the attention of researchers recently. Specifically, MNBs have been shown to have a significantly lower rising velocity than larger bubbles, and their negatively charged surface prevents coalescence, so they may persist in water for longer periods [24]. MNBs also have an internal pressure and a specific surface that favors mass transfer, increasing ozone dissolution. In addition, the effective generation of $\bullet$OH radicals by the collapse of $O_3/MNBs$ under a broad pH band can contribute to the oxidation of organic molecules [25–27], increasing the effectiveness with respect to conventional $O_3$ processes. As an example, Azuma et al., 2019, reported an increase in the rates of pharmaceutical compound removal between 8% and 34% when a combined $O_3/MNBs$ treatment was used, as compared with $O_3$ alone. Thus, the degradation rates of compounds such as acetaminophen glucuronide, bortezomib acid, or loxoprofen, increased from 19% to 32%, 48% to 88%, and 48% to 82%, respectively, while less persistent compounds such as carbamazepine or acetaminophen were completely removed [28].

However, despite all these promising advantages, most studies have been conducted at the laboratory scale and under controlled conditions, or even using distilled or simulated water as a model. Therefore, studies on the benefits of using $O_3$/MNBs systems using real wastewater are limited, and even more so, the application of this technology on a large scale. Consequently, factors such as design parameters for large-scale reactors, their operation, process control, or associated costs are not yet sufficiently studied. In addition, there is limited evidence on the benefits for disinfection purposes. Another uncertainty comes from the lack of knowledge about the transformation products (TPs) related to the process of degradation of pharmaceuticals that could be generated in this type of system and their impact on the environment.

This work aims to evaluate the combined use of MNBs/$O_3$ in a large-scale regeneration system to obtain water of high agronomic quality. Pharmaceutical removal efficiency, including COVID-19-related compounds, along with disinfection, as well as transformation products generated and cost evaluation, will be key aspects in system optimization and evaluation.

## 2. Results and Discussion

### 2.1. Pharmaceuticals Removal Efficiency

The results provided evidence of the influence of the workflow on pharmaceutical removal percentages, showing higher efficiencies working at 2000 L/h and 500 L/h as compared to the intermediate flow rate (1200 L/h) (see Figure 1). This phenomenon can be due to two different issues: the dilution carried out and, therefore, the final concentration of ozone, and the reaction with molecular ozone. At a high workflow, more turbulence is generated in the wastewater, so the MNBs collide with each other and with $O_3$ molecules, increasing the concentration of reactive species (mainly $^\bullet$OH radicals) capable of reacting and breaking down pharmaceuticals [29]. Since $^\bullet$OH has a higher oxidation potential than $O_3$, the pharmaceuticals oxidation process will increase with respect to a lower workflow [30]. On the other hand, at low working flows, the $O_3$-MNBs/pharmaceuticals ratio is higher, meaning a higher ozone concentration which increases the oxidizing species, favoring the removal of pharmaceuticals [31].

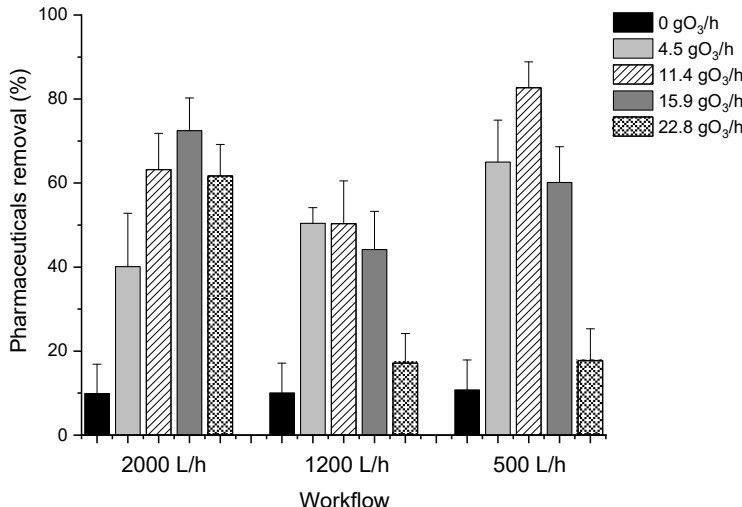

**Figure 1.** Influence of inlet ozone concentration and flow rate in pharmaceuticals removal.

A clear relationship was also observed between the pharmaceutical removal percentages and ozone production, according to Dawood et al., 2021 [29]. In particular, an increase in the ozone dose applied was related to an increase in pharmaceuticals removal, reaching maximum values of 72.5 ± 7.8% working with 15.9 g$O_3$/h, 50.3 ± 10.2% working with 11.4 g$O_3$/h and 82.7 ± 6.2% working with 11.4 g$O_3$/h, when 2000 L/h, 1200 L/h, and 500 L/h workflow were tested, respectively. However, a decrease in removal efficiency was observed in all cases when working at maximum ozone production values (22.8 g$O_3$/h).

These effects may be related to ozone solubility. In general, the increase in ozone concentration promotes the transfer of ozone from the gas phase to the liquid phase up to the solubility limit. At this point, there is no transfer of a greater supply of ozone to the liquid phase, so the efficiency of pharmaceutical removal remains constant or even decreases, according to Joseph et al. (2021) [32]. This was tested experimentally by measuring dissolved oxygen (indigo method). In all cases, the dissolved ozone percentage increased as the inlet ozone concentration increased, reaching a maximum value of 82.0% working with 15.9 $gO_3$/h, a dissolved ozone percentage similar to those reported by Kokkoli et al. (2021) [33]. From that point on, this percentage remained constant and even decreased, thus corroborating our hypothesis.

On the other hand, it should be noted that, in the absence of ozone, an average removal of 10% of pharmaceuticals was observed, regardless of the workflow used, suggesting the ability of MNBs to enhance the efficiency of the process. These results are consistent with the experiments by Bui et al. (2020) [34]. These authors reported that the electrostatic attraction between MNBs and organic matter (due to the opposite surface charge between molecules), along with reactive species that can be generated by the collapse of the MNBs, lead to a decrease in organic matter in wastewater.

The percentages obtained from Figure 1 were also related to the ozone consumed in the process. The results (Figure 2) showed that the highest ozone consumption occurred at 500 L/h, followed by 1200 L/h and 2000 L/h, respectively. This allowed us to corroborate the hypotheses and justify the results presented above. In this sense, at high flows, indirect $O_3$ reactions would predominate, due to the presence of a greater number of radical species ($^\bullet$OH), due to the reactivity of the MNBs (molecular collision). However, at low flows, direct $O_3$ reactions may predominate. In both cases, the reactivity would be greater than at intermediate flows, where the contact time between molecules is lower, and there is not enough flow to make collisions between molecules too effective, so there would be a balance between direct and indirect reactions, limiting them both. On the other hand, a decrease in ozone consumption was observed at the highest ozone generation rate, verifying that this concentration reaches the $O_3$ solubility limit.

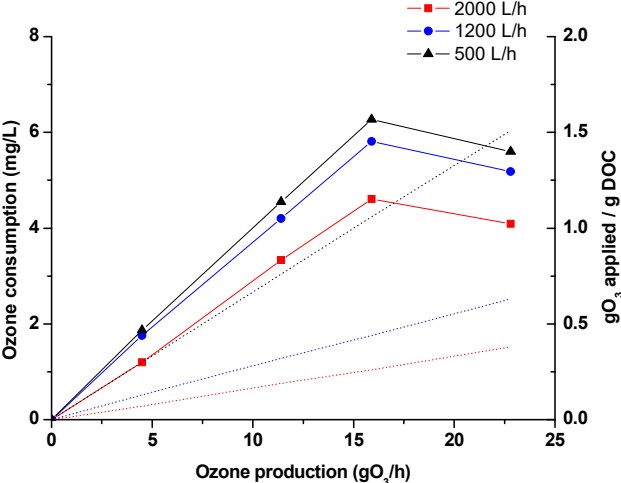

**Figure 2.** Ozone consumption (straight line) and applied normalized ozone (dash line) under experimental conditions.

On the other hand, the specific ozone dose applied, normalized to the dissolved organic carbon (DOC) of effluents, showed values between 0 to 0.38 $gO_3$/gDOC, 0 to 0.63 $gO_3$/gDOC, and 0 to 1.51 $gO_3$/gDOC when flowrates of 2000 L/h, 1200 L/h, and 500 L/h were used, respectively. In all cases, the values of $gO_3$ applied/g DOC working at 2000 L/h were much lower than those reported in the literature for conventional $O_3$ reactions in large-scale systems. Specifically, Hollender et al. (2009) studied the removal of 24 pharmaceuticals in wastewater with $O_3$ at full scale. The results showed that a dose of

0.47 gO$_3$/gDOC was needed to remove compounds of greater reactivity (including CBZ or SMX), while higher doses were needed for slow-reacting compounds [35]. The same removal of compounds under conventional O$_3$ reactions was studied by Antoniou et al. (2013) [36], demonstrating that a dose between 0.55 gO$_3$/gDOC to 0.77 gO$_3$/gDOC was necessary to reach a 90% removal of highly reactive compounds while for slow-reacting compounds (such as NPX) a specific ozone dose > 1 g O$_3$/gDOC was necessary. Therefore, the positive impact of the use of MNBs in O$_3$ reactions is demonstrated, decreasing the gO$_3$/gDOC to be applied as compared to conventional ozonation reactions, which could also have an impact on cost reduction.

According to these results, and considering that the treated water is intended to be reused for agriculture irrigation, with disinfection crucial for optimization, an evaluation of disinfection based on *E. coli* removal during the process under the selected operational conditions was performed. The results (Figure 3) showed that the reduction in *E. coli* bacteria was significantly affected by ozone dosage and workflow, as was the case for pharmaceutical removal, with the mechanisms being the same. A combination of direct (involving O$_3$ molecules) and indirect (involving free $^\bullet$OH) reactions are involved in pharmaceuticals removal and *E. coli* inactivation processes under selected wastewaters due to their pH value (~pH 7), and this reactivity is further enhanced by the presence and behavior of MNBs under particular conditions [37–39]. Therefore, the study of the operating conditions for each specific system is essential for achieving the highest removal efficiency.

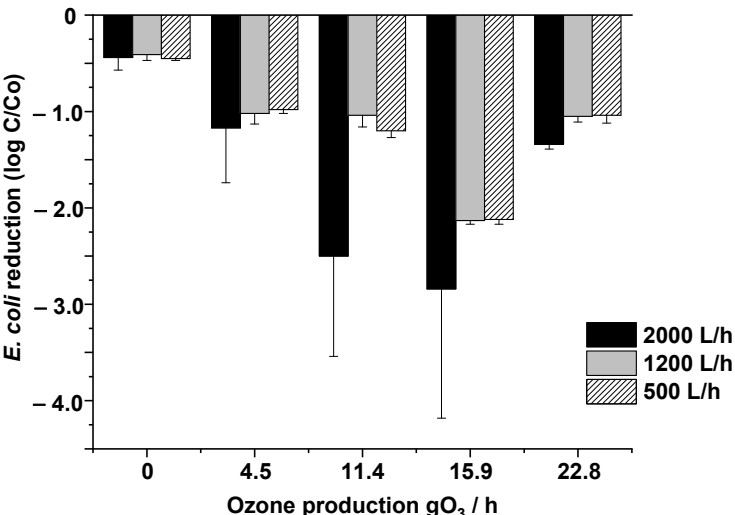

**Figure 3.** Influence of inlet ozone concentration and flow rate on *E. coli* removal.

Considering that the initial abundance of *E. coli* in the inlet wastewater was 2.3 × 10$^4$ ± 1.1 × 10$^4$ CFU/100 mL (see Table S1), 15.9 gO$_3$/h of ozone production was the only condition able to inactivate *E. Coli* values below LOD, showing reduction values of up to 4-log, while lower removal efficiencies were obtained for the other conditions tested. In all cases, the best removal efficiencies were observed using the 2000 L/h workflow, due to increased reactivity as a result of the collisions between molecules, while the use of air MNBs managed to remove an average of 0.5 log, demonstrating that the hydroxyl radicals alone, generated by the collapse of the MNBs, have a disinfectant activity, according to Sumikura et al., 2007 [40]. In this work, the authors reported a 24.5% removal of *E. coli* in wastewater using micro-air bubbles, while this percentage increased progressively when increasing amounts of ozone were added.

Therefore, considering the combination of the efficient disposal of pharmaceuticals and disinfection requirements for agricultural reuse, 2000 L/h and 15.9 gO$_3$/h were considered optimal conditions for continued work in the large-scale system.

## 2.2. Prediction of the Pharmaceutical Removal from Physico-Chemical Properties

Despite the high efficiencies in pharmaceutical removal obtained under the different operational conditions studied, not all compounds behaved in the same way in the $O_3$/MNBs system. Specifically, of the 12 pharmaceuticals analyzed, 6 of them showed high removal percentages, reaching, in all cases, values higher than% (TRZ, TCL, AMX, CBZ, ACT, and SMX), 4 of them (KTP, NPX, DCF, and ERY) showed moderate efficiencies (between 57.62% and 77.34%), while the remaining 2 (HLP and CHL) had low removal efficiencies (approximately 22.50%, see Figure 4). This may be related to the structure and functional groups of the pharmaceuticals to be degraded and, therefore, to the reactivity of these groups with $O_3$ and $^\bullet$OH radicals in an aqueous solution, and may result in oxidation processes, cycloadditions, or mainly electrophilic substitution reactions according to Beltran et al., 2003 [41]. Specifically, the presence of electron-rich functional groups has been reported to react with $O_3$ molecules by electrophilic substitution via electron transfer, increasing removal percentages [42]. These groups include C=C bonds (mainly found in CBZ), nitrogen-containing compounds (mainly found in TRZ, TCL, or SMX), or organosulfur compounds such AMX. However, electron-withdrawing functional groups, such as fluoro, chloro, or carboxyl groups (found in DCF, CHL, or HLP) reduce electronic density from pharmaceuticals inhibiting electrophilic substitution reactions, resulting in a shielding effect [36]. In this perspective, and in order to simplify and predict the pharmaceutical removal efficiency in $O_3$/MNBs systems, the removal percentages obtained were directly associated with the ionic capacity of the pharmaceuticals, following the trend: cationic compounds > dipolar ions > neutral compounds > anionic compounds. Thus, cationic, dipolar ions, and neutral compounds would be considered as high removal rate compounds, while the anionic compounds would be considered as compounds with moderate removal percentages. However, CHL and HPL, despite being cationic compounds, showed the lowest removal rates (22.5 ± 2.6% and 22.5 ± 3.1%, respectively). This behavior could be associated with the presence of electron-withdrawing functional groups (mainly fluoro and choro in their structure, limiting their removal [43]. On the other hand, another exception to this behavior was the case of ERY. Despite being a neutral compound, its low reactivity was due to the scarcity of its C=C bond and other electron-rich functional groups in their structure, which limited its reactivity. In this sense, knowing the polarity and chemical structure of the pharmaceuticals to be degraded by $O_3$/MNBs reactions could be crucial for predicting the removal efficiency of selected compounds in real systems.

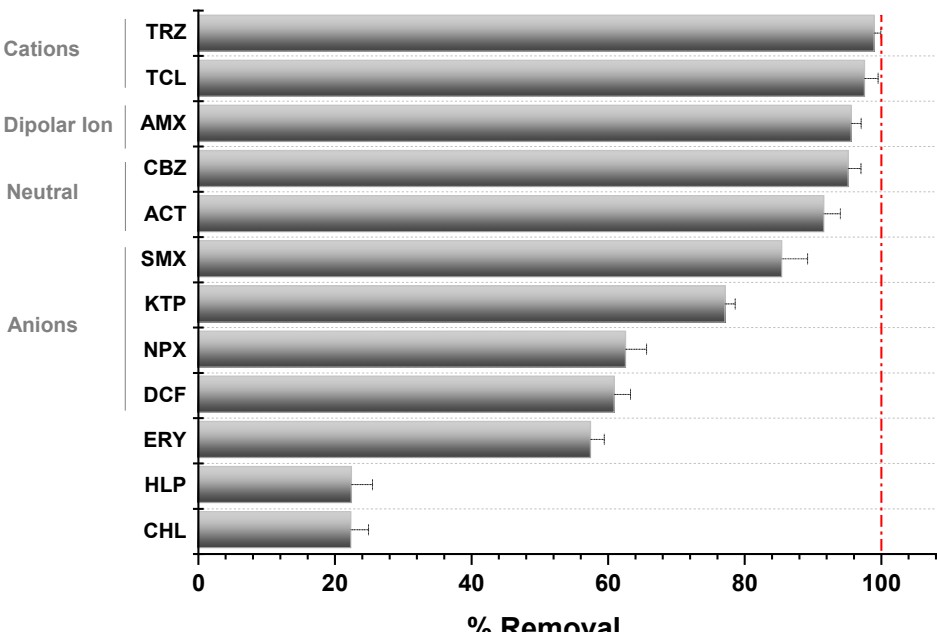

**Figure 4.** Influence of ionic capacity in pharmaceuticals removal.

### 2.3. Transformation Products Generated in the Process

One aspect to consider for the implementation of $O_3$/MNBs-based systems on a large scale is related to the TPs that can be generated due to the degradation of pharmaceutical compounds during the process due to $^\bullet$OH radical reactivity [44]. Although the generation of TPs in ozonation-based processes ($O_3$, $O_3/H_2O_2$, etc.) and their potential effect on the environment have been studied for many years [45], to our knowledge, there are no studies yet that have considered the TPs generated in $O_3$/MNBs processes. In this light, a study was conducted to evaluate whether the TPs generated in $O_3$/MNBs processes are consistent with those reported in the literature for other $O_3$-based processes [46–55].

Of the 104 TPs considered in the suspect list, only 9 chromatographic peaks were detected as positive, following the settings described in Section 2.3. After grouping retention time signals, only five remained, which originated from CBZ (three TPs (TPC10, TPC4, and TPC13) at retention times of 6.54, 8.67, and 9.45 min, respectively), DCF (1 by-product at a retention time of 9.08 min, TPD6), and NPX (1 by-product at a retention time of 10.83 min, TPN1), with the proposed structure previously reported in the literature (see Table S2). For all selected chromatographic peaks, score values were 100% and the mass error was <5 ppm. For more information, see Table S3.

Significant differences were observed according to the different workflows. In particular, TPC4 (carbamazepine by-product) was the largest by-product area and was detected only at the 2000 L/h (see Figure 5). It formed from 4.5 g$O_3$/h and the peak area increased following the ozone production, while a small decrease was detected at 22.8 g$O_3$/h, following a pattern similar to the pharmaceutical's behaviour. Although several authors have found this by-product of carbamazepine in ozonation reactions, and some of them have reported it as one of the most persistent TPs [56,57], it was Kråkström et al. (2020) [51] who first associated this mass to a specific structure (2,2'-azanediyldibenzaldehyde) using NMR. The authors reported that this by-product was formed via radical reactions from the intermediate product N, N-bis(2-formylphenyl) urea, continuing to react with $^\bullet$OH radicals forming other intermediates, confirming their quick reactivity under higher flow rates.

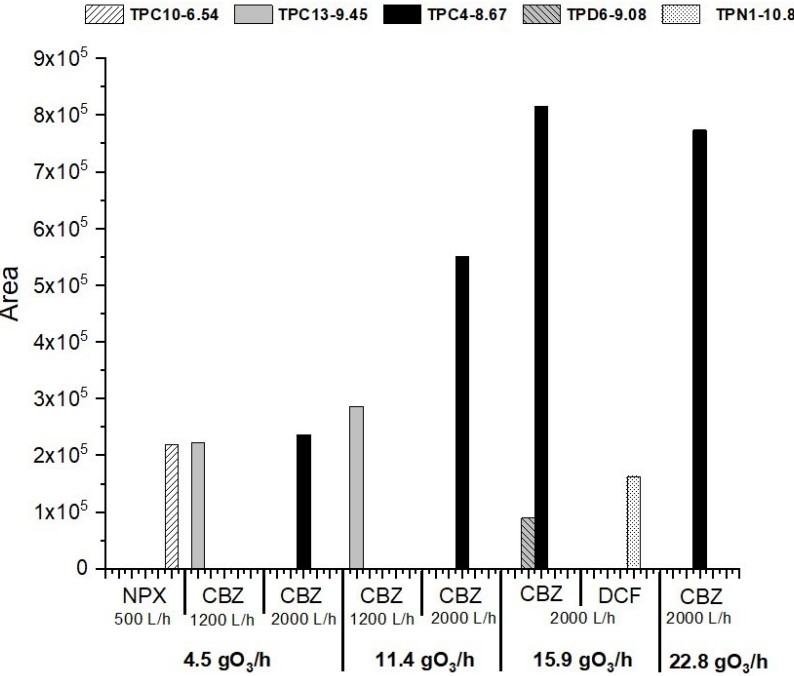

**Figure 5.** TPs obtained according to the selected suspect list under different operational conditions.

On the other hand, TPC10 was detected in the 1200 L/h workflow under 4.5 and 11.4 g$O_3$/h ozone productions, while at higher production values, it was not detected, probably due to its reactivity under the $O_3$/MNBs system. Although it is difficult to predict

their origin, these TPs could be hydroxylated derivatives formed from other carbamazepine TPs, according to Kråkström et al. (2020) [51]. Other TPs, such as TPN1, TPC13, or TPD6 were found on a spot basis in the samples, which shows that, although they can be formed in $O_3$-MNBs, their persistence is lower than the other TPs found.

However, despite the preliminary results obtained, the appearance of other TPs cannot be ruled out if the concentrations of the starting pharmaceuticals were higher, as many studies reported in the literature were carried out with concentrations close to 1000 µg/L, five times higher than the concentration tested in this study.

*2.4. Agronomic Quality of System Effluents under Optimal Operation Conditions*

In order to determine whether the effluents from the proposed and optimized full-scale $O_3$/MNBs system would be suitable for agricultural use, the most recent EU directive (EU, 2020/741) on minimum requirements for water reuse was chosen as a reference [15]. Particularly, this directive stipulates four different "classes of reclaimed water quality" (A, B, C, and D) based on permitted agricultural use and irrigation method, with class A being the most restrictive. With this aim, the system with real effluents (non-spiked wastewater) under optimal operating conditions (2000 L/h and 15.9 $gO_3$/h) was monitored working in continuous mode for 6 months (2 monthly samples).

A significant reduction in the different physico-chemical parameters was detected in all $O_3$/MNBs effluents analyzed. Specifically, turbidity, $BOD_5$, and TSS values decreased by 45.0 ± 10.5%, 33.3 ± 15.2%, and 15.1 ± 2.0%, respectively, showing in all cases values below the most restrictive limits included the EU regulation (class A) ($BOD_5 \leq 10$ mg/L; TSS $\leq 10$ mg/L and turbidity $\leq 5$ NTU). On the other hand, an increase of approximately 9% in transmittance values was detected in the system effluents, from 53.7 ± 4.6% to 62.4 ± 3.3%, while the pH and ion content remained practically constant, with pH values of 7.4 ± 2.1 and sodium adsorption ratio values of less than 6 meq/L, with the limit regulated for agricultural practices being 6 meq/L according to Spanish Royal Decree 1620/2007 [58].

Regarding the pharmaceuticals, all the samples entering the system showed low concentrations of all analyzed compounds, with average values between 0.03 µg/L and 0.08 µg/L, mainly because the WWTP used for the study was located in a rural area with an agricultural perspective, so the concentration of pharmaceuticals that reached this WWTP was relatively low throughout the year. In spite of this, all the samples treated using the $O_3$/MNBs system showed a total removal of pharmaceuticals, with values, in all cases below the analytical detection limits.

As for the disinfection purposes, the effluents showed a significant reduction in *E. coli* (between 3-log to 4-log), with concentrations in all cases below the European regulation on minimum quality requirements established for this bacterium (<10 CFU/100 mL), the value marked for "the point of compliance" in "class A" reuse. Recently, several authors have demonstrated the potential of $O_3$/MNBs combination for wastewater disinfection. Specifically, Furuichi et al. (2013) reported the inactivation of Gram-positive and Gram-negative bacteria, including *E. coli*, when a combination 1.5 mg/L of dissolved ozone and MNBs were used as the water treatment obtained removal percentages higher than 99.99% (from $1.6 \times 10^9$ CFU/mL to <10 CFU/mL) [59], while Cruz et al., 2017 [60], analyzed the inactivation of fecal and total coliforms in domestic wastewaters from Peru, demonstrating the reduction of more than 99% for both indicators.

However, despite the promising results, studies evaluating the efficiency of $O_3$/MNBs systems in the inactivation of highly persistent microorganisms are limited. Under this perspective, an additional study based on the removal efficiency of *Clostridium perfringens spores* as protozoa indicator was carried out. This indicator was selected due to their recent introduction in EU 2020/741 and their resistance to complete removal in conventional tertiary treatments (including chlorine, UV-C light, or even conventional ozonation) as compared with other parasites [61–63]. The results showed an average removal percentage of 75 ± 10% during continuous work under optimal conditions (from 800 ± 141 CFU/100 mL in the $O_3$/MNBs inlet wastewater). However, water quality changes in *Clostridium perfringens*

*spores* removal were detected during storage of treated water, reaching 90% to 99% removal efficiencies after 24–72 h. This may be related to the persistence of MNBs in water, lengthening the activity of ozone and radical species in the solution, thus prolonging the antimicrobial activity [64]. These changes in water quality during storage have been reported by other authors. Chuajedton et al., 2016 [65] studied the use of a lab-scale $O_3$/MNBs system for the inactivation of *E. coli* in fresh-cut pineapple. The authors obtained the highest *E. coli* removal values (between 4–5 log reduction) after 2 days of storage. On the other hand, Seki et al., 2017, studied the microbicidal effects of $O_3$/MNBs after long-term storage. The results confirmed the ozone retention capacity in the MNBs, and their antimicrobial activity, removing resistant microorganisms even after long storage seasons under suitable conditions [66].

*2.5. Cost Assessment Approach*

Since the removal of pharmaceutical compounds under advanced tertiary treatments (including $O_3$/MNBs) are reported as energy-intensive processes, the energy requirement is one of the most important factors to consider from an economic point of view [67,68]. Under this perspective, an estimation of the economic cost of the process based on electrical energy consumption is presented. The analysis was performed according to figures of merit related to $^\bullet OH$-involved processes described by the International Union of Pure and Applied Chemistry (IUPAC) [69]. Following the marked recommendations, the electrical energy per order factor ($E_{EO}$) was calculated. This factor describes the electric energy (kWh) per unit of volume ($m^3$ of wastewater) required to degrade a pharmaceutical compound by one order of magnitude. $E_{EO}$ can be calculated with the following equation:

$$E_{EO} = \frac{P}{F \times \log(Ci/Cf)} \tag{1}$$

where P is the rated power (kWh) of the oxidation system, F is the flow rate ($m^3$) and Ci and Cf are the initial and final concentrations of the pharmaceutical compounds [70]. The energy consumption of pumping, ozone generation, and MNBs formation was found to be about 6.1 kWh. Considering that $E_{EO}$ values are directly related to the pharmaceutical removal efficiency (log Ci/Cf), two different groups were considered: group 1 (99% average removal efficiencies, such as some cations) and group 2 (90% average removal efficiencies, mainly dipolar ions or some neutral compounds). The calculated $E_{EO}$ values were 1.80 kWh/$m^3$/order for group 1 and 3.05 kWh/$m^3$/order for group 2, for the $O_3$/MNBs process. To our knowledge, this is the first time that the costs associated with a large-scale $O_3$/MNBs process have been calculated, and these results confirm that the energy costs are directly associated with the physico-chemical properties of the compounds to be degraded. Therefore, in WWTP where group 1 compounds predominate, costs would be lower as compared to other WWTP having compounds of higher persistence. The calculated $E_{EO}$ values were analogous to those predicted by Miklos et al. (2018) and Joseph et al. (2021) [32,71] for the implementation of large-scale ozonation systems. However, it would be necessary to specifically know the energy requirements that were considered for these predictions (such as pumping), as well as redesign the location of the tertiary step in the treatment line under the criteria of energy optimization. At any rate, $E_{EO}$ values were in line with those predicted by the same authors for other advanced oxidation processes. Specifically, the authors suggested an average energy consumption for conventional large-scale ozonation processes of 1 kWh/$m^3$/order, while higher values (between 2.6 (photo-Fenton) to 38.1 kWh/$m^3$/order (eAOP)) were suggested for the degradation of specific pollutants with AOPs technologies. On the other hand, studies reported higher $E_{EO}$ values for UV-based photocatalysis (335 kWh/$m^3$/order), ultrasound (2616 kWh/$m^3$/order), or microwave-based AOPs (543 kWh/$m^3$/order), so the use of large-scale $O_3$/MNBs could have great energy benefits over other technologies. However, despite these approaches, there is a great lack of knowledge about the energy costs required in $O_3$/MNBs large-scale systems, as $E_{EO}$ values are mostly dependent on experimental conditions (pH, physico-chemical properties of the water to be treated, etc.), mainly due to the competition between direct and

indirect ozonation reactions. In spite of this, the advantages of using $O_3$/MNBs systems are promising. Azuma et al. reported that the combination of $O_3$/MNBs could reduce energy costs as compared to conventional $O_3$ treatments due to the higher consumption of $O_3$ during the reaction and its rapid diffusion into the liquid phase [28].

On the other hand, it is important to note that the estimated $E_E O$ values reported generally do not reflect the additional energy demand required for the production of chemicals or catalysts used in processes, so these factors should be included in the $E_E O$ calculation [72]. As such, when evaluating the overall costs, e.g., when calculating capital expenditures (CapEx) and operational expenditures (OpEx), other costs such as chemicals and material cost, control, and maintenance expenses should be included and determined. Nevertheless, $E_E O$ is a good starting point for comparing the energy requirement of ozone for pharmaceutical removal.

## 3. Materials and Methods

### 3.1. System Design

The combined $O_3$/MNBs full-scale system (Figure 6a) was a non-commercial system specifically designed by SISTEMA AZUD, S. A. for these tests. It was installed in a conventional WWTP located in Murcia Region (Spain) (latitude 37°47′48″ N, longitude 0°57′36″ W). Specifically, this WWTP receives domestic and industrial effluents from three municipalities (Roldán, Lo Ferro, and Balsicas) and has a maximum treatment capacity of 2,007,500 m³/year of wastewater. The water line consists of a primary treatment, followed by an activated sludge secondary treatment with extended aeration and sedimentation, and a final tertiary treatment based on coagulation–flocculation, sand filter, and disinfection using UV radiation. WWTP effluents are commonly used for agricultural irrigation.

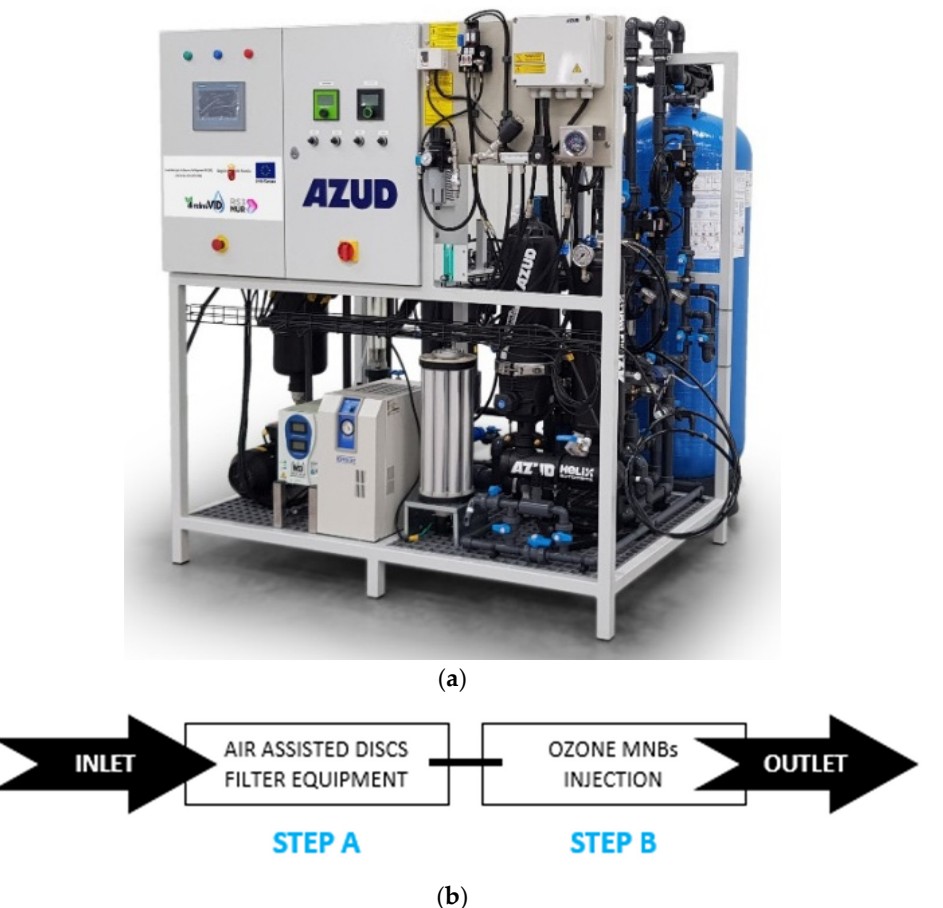

(**a**)

(**b**)

**Figure 6.** (**a**) $O_3$/MNBs system design; (**b**) system treatment line steps.

The designed system was connected to the WWTP secondary clarifier, bypassing the WWTP's conventional tertiary treatment. The physico-chemical characterization of inlet wastewater is reported in Table S1. The system consists of a modular and compact system with combinable steps, automation for monitoring operational parameters, and remote control. The workflow range during operation is 500 L/h to 2000 L/h.

The experimental treatment line is based on two different stages arranged in series (Figure 6b). Both stages treat the entire treatment flow, so there is no shunt in which the treated flow can be diluted.

Step A: The first stage is an air-assisted disc filtering equipment AZUD HELIX AUTOMATIC®. Specifically, the experimental module is based on a mono stage equipment, AZUD HELIX AUTOMATIC FT201/2VX AA DLP (Sistema AZUD, S.A., Murcia, Spain), provided by a filtration grade limit of 50 μm. The disc filter technology was selected because it allows deep filtration that removes all particles larger than the filtration grade and a high percentage of smaller ones, regardless of geometry or nature, protecting the next step, according to Ponce-Robles et al., 2020 [73]. In addition, this experimental module includes an automatic washing system that combines water and compressed air, which allows working at low pressure (even during the counter-washing process), reducing the hydraulic interaction of the system during cleaning, optimizing water and energy consumption, and achieving total disk recovery [73].

Step B. The second stage is based on ozone micro-nano bubbles (MNBs) injection. This step consists of two combined systems: (i) ozone generation system; (ii) ozone mass transfer system based on MNB technology. The ozone generator system is fed by on-site-treated atmospheric air. The air treatment line is composed of an air dryer, a particle filter, and an oxygen concentrator. The percentage of oxygen in the air after this treatment can be increased up to 90%. The ozone generation system, which uses corona discharge technology, is fed at 5 lpm of this oxygen-concentrated air to produce a maximum of 22,800 $mgO_3$/h. Likewise, the generation can be regulated between 0–100%. The ozone mass transfer system consists of a non-commercial prototype composed of an MNBs bubble injector which is simultaneously fed by the ozone generator and filtered water. This system is based on a gas–liquid mixing pump that absorbs the gas phase by negative pressure on the suction inlet and has a high-speed rotary impeller to stir liquid and gas phases.

An optional ozonization tank is located at the outlet of the $O_3$/MNBs system to study the behavior in water after an established period. The volume of this tank is 5.000 L.

### 3.2. Pharmaceutical Compounds Included in the Study

A total of 12 pharmaceutical compounds were selected as a case study considering different aspects: (i) Cover pharmaceuticals from different therapeutic groups (analgesics, antibacterial, anticonvulsants, antimalarials, anti-inflammatories, antipsychotics, antidepressants); (ii) Study of persistent or bio-recalcitrant compounds commonly found in worldwide WWTP effluents, such as acetaminophen (ACT), amoxicillin (AMX), carbamazepine (CBZ), diclofenac (DCF), erythromycin (ERY), ketoprofen (KTP), naproxen (NPX), sulfamethoxazole (SMX), or tetracycline (TCL) [4]; (iii) Study of pharmaceuticals included in the list of trial drugs for the diagnosis and treatment of COVID-19, and so, considered essential for the management of sanitary emergence, such as chloroquine (CHL) [74], trazodone (TRZ), [75] or haloperidol (HLP), (included in the Resolution from 19 June 2020, of the Spanish Agency for Medicines and Medical Devices, on urgent prevention, containment, and coordination measures to address the health crisis caused by COVID-19) [76].

Pure standards (>98%) of selected compounds were purchased from Sigma–Aldrich® (Steinheim, Germany). Individual stock standard solutions and work solutions used in both analytical and experimental viewpoints were prepared in methanol or MilliQ water and stored at −20 °C. The physico-chemical properties of all selected compounds are reported in Table 1 [77,78].

**Table 1.** Physico-chemical properties of selected pharmaceuticals.

| | Drug Type | Formula | Molecular Weight (g/mol) | Log Kow | pKa |
|---|---|---|---|---|---|
| Acetaminophen (ACT) | Analgesic | $C_{37}H_{67}NO_{13}$ | 151.16 | 3.06 | 8.9 |
| Amoxicillin (AMX) | Antibacterial | $C_{16}H_{19}N_3O_5S$ | 365.4 | 0.87 | 2.6 |
| Carbamazepine (CBZ) | Anticonvulsant | $C_{15}H_{12}N_2O$ | 236.3 | 2.45 | 15.9, −3.8 |
| Chloroquine (CHL) | Antimalarial | $C_{18}H_{26}ClN_3$ | 319.9 | 4.63 | 10.1 |
| Diclofenac (DCF) | Anti-inflammatory | $C_{14}H_{11}C_{l2}NO_2$ | 296.1 | 4.51 | 3.9 |
| Erythromycin (ERY) | Antibacterial | $C_{37}H_{67}NO_{13}$ | 733.9 | 3.06 | 8.8 |
| Haloperidol (HLP) | Antipsychotic | $C_{21}H_{23}ClFNO_2$ | 375.9 | 4.30 | 8.65 |
| Ketoprofen (KTP) | Anti-inflammatory | $C_{16}H_{14}O_3$ | 254.3 | - | 3.98 |
| Naproxen (NPX) | Anti-inflammatory | $C_{14}H_{14}O_3$ | 230.26 | 3.18 | 4.18 |
| Sulfamethoxazole (SMX) | Anticonvulsant | $C_{10}H_{11}N_3O_3S$ | 253.28 | 0.89 | pKa1 = 1.6 pKa2 = 5.7 |
| Tetracycline (TCL) | Antibacterial | $C_{22}H_{24}N_2O_8$ | 444.4 | −1.37 | 7.68, 3.3 |
| Trazodone (TRZ) | Antidepressant | $C_{19}H_{22}ClN_5O$ | 371.9 | 3.21 | 6.79 |

Data from PubChem (https://pubchem.ncbi.nlm.nih.gov/ (accessed on 1 December 2022)) and network of reference laboratories, research centers, and related organizations for monitoring of emerging environmental substances (https://www.norman-network.net/ (accessed on 1 December 2022)).

### 3.3. Analytical Determinations

Total suspended solids (TSS), electrical conductivity (EC), pH, turbidity, dissolved organic carbon (COD), 5-day biological demand ($BOD_5$), total nitrogen (TN), and chemical oxygen demand (COD) analyses were carried out following Standard Methods (APHA, 2012) [79].

The dissolved ozone in the system was determined using the indigo method according to the standard methods (APHA-AWWA-WEF, 1992) procedure. The analysis was performed using potassium indigo tri-sulfonate (Sigma–Aldrich®, Steinheim, Germany) and an ultraviolet-visible (UV-vis) spectrophotometer set at 600 nm [80].

*Escherichia coli* measurements were determined by the standard membrane filtration method, according to ISO 9308-1:2014 [81]. The detection limit was set at 1 CFU/ 100 mL (Colony Forming Unit per 100 mL), according to the Class A maximum value (10 CFU of *E. coli*/100 mL) set by the new European Regulation on minimum requirements for water reuse ((EU) 2020/741). *Clostridium perfringens* spores were analyzed through an external ENAC-certified reference laboratory (IPROMA S.L) located in Castellón, Spain, following the standard membrane filtration method established in the UNE-EN ISO 14189:2007 [82], based on selective cultivation on TSCF agar at 44 ± 1 °C for 21 ± 3 h and subsequent colony identification by acid phosphatase reaction. The limit of detection was set at 1 CFU/100 mL.

A liquid chromatography-quadrupole time-of-flight mass spectrometry (UPLC-QqTOF-MS) system was used for targeted and non-targeted analysis (pharmaceuticals and TPs). The LC separation was achieved using an ACQUITY I-Class UPLC system (Waters Corporation, Milford, MA, USA) coupled with an ACQUITY BEH C18 (100 mm × 2.1 mm, 1.7 µm) column, according to Martínez-Alcalá et al., 2017 [83]. The LC system was connected to a Bruker Daltonics, maXis q-TOF mass spectrometer equipped with an electrospray ion source (Bruker Daltonics, Bremen, Germany). The system worked via a TOF-MS survey scan (resolving power ≥ 55,000 FWHM). The targeted and non-targeted substances were

identified and reported from accurate-mass scan data using the software Target Analysis (1.3) and Data Analysis (4.2) from Bruker. For more information, see general information S1. Before injection into the chromatographic system, samples were filtered using a 0.22 μm PTFE filter (Millipore). The detection limit (DL) for the selected compounds was adjusted to 5 μg/L with an associated error for each concentration level < 10%, using matrix calibration curves. An additional sample pre-concentration step was performed in real samples containing pharmaceutical compounds below limits set using solid phase extraction (SPE), allowing for a decrease in the detection limit by two orders of magnitude (detailed SPE procedure is described in general information S2).

For TP analysis, the literature research focused on previously reported TPs under $O_3$ wastewater treatments. The suspect list developed contained 104 TPs (see Table S2), and included transformation products of TRZ, TCL, AMX, CBZ, SMX, KTP, NPX, DCF, and ERY, while TPs of other compounds such as ACT, HPL, CHL, or TRZ were excluded since relevant $O_3$ TPs were not found in the literature [46–55].

Only chromatographic peaks with an absolute intensity threshold of 1000 counts per second (cps) and an exact mass matching the values reported in the literature were considered tentative candidates. Finally, only candidates with an intensity response 10 times higher than the analyzed blanks [84] and showing score values close to 100% (intrinsic statistical parameter of the chromatographic peak quality indicator system) were considered positive.

*3.4. Experimental Procedure and Sampling*

All experiments were performed in the full-scale system in continuous mode at ambient temperature (18.0 ± 5.0 °C) and natural wastewater pH (7.4 ± 0.2).

Different percentages of ozone generation, together with flow rate effects on pharmaceutical removal efficiency, were studied to analyze different operating conditions and optimize the proposed system. With this aim, three working flows (i) 2000 L/h; (ii) 1200 L/h and (iii) 500 L/h of inlet treated wastewater, and 5 different percentages of ozone production, adjustable in the ozone generator (i) 0%, corresponding only to the injection of purified atmospheric air; (ii) 25%; (iii) 50%; (iv) 75%; (v) 100% were studied. These production percentages are related to (i) 0.0; (ii) 4.5; (iii) 11.4; (iv) 15.9; (v) 22.8 $gO_3$/h of ozone produced, respectively. To ensure a constant ozone production rate throughout the experiment, all samples were collected 15 min after ozone generator activation and MNBs injection. All optimization experiments were performed using real secondary effluents spiked with 200 μg/L of selected pharmaceuticals. For this, required volumes of a stock solution containing pharmaceutical compounds were added to real wastewater into an external reaction tank connected directly to the system, as pharmaceuticals are commonly found in real wastewater in the ng/g range [4]. This initial concentration was chosen as it is sufficiently high to obtain degradation values with available analytical techniques and low enough to simulate real environmental conditions. The experiments were conducted in triplicate to ensure repeatability, and the averaged data are presented below. All samples were collected in 1-L amber glass bottles and taken directly to the laboratory for further analysis (stored at 4 °C in the dark and analyzed within 24 h).

Once the optimization phase was completed, the efficiency of the system was monitored under real-world conditions (real unfortified water) following the same sampling and analysis protocols. Finally, an evaluation of the TPs likely to be generated in the system was carried out, as well as a detailed study of the associated costs.

## 4. Conclusions and Future Perspectives

This work clearly demonstrates the feasibility of implementing $O_3$/MNBs systems as a tertiary treatment in conventional WWTPs for agronomical purposes. After optimization, the workflow of 2000 L/h and ozone production of 15.9 $gO_3$/h showed the best results, with pharmaceutical removal percentages of 72.46 ± 7.8% and a significant *E. Coli* reduction between 3-log and 4-log.

The presence of MNBs favored the efficiency of the system, improving the direct (involving $O_3$ molecules) and indirect (involving free $^\bullet OH$) oxidation mechanisms. However, in low workflows, the direct reaction predominates, while in high workflows, the indirect reaction predominates, due to the reactivity of the MNBs (molecular collision).

As for pharmaceutical removal percentages, three ranges can be considered: removal efficiencies higher than 85.68% (cationic, dipolar ions, and neutral compounds), moderate removal efficiencies, between 57.62% and 77.34% (anionic compounds), and low removal efficiencies of 22.50% (pharmaceuticals that contain mainly choro and fluor groups in their structure). In this sense, knowing the polarity and chemical structure of the pharmaceuticals to be degraded by $O_3$/MNBs reactions could be crucial for predicting the removal efficiency of the selected compounds in real systems.

The energy consumption required was also associated with the structure of the pharmaceutical compounds. Therefore, in WWTPs where cationic groups predominate, the costs would be lower as compared to other WWTPs that are more persistent.

The effluents generated in the system showed high agronomic quality, complying with the standards set in the European regulation for wastewater reuse. In addition, quality changes were detected during storage, increasing disinfection after 24–72 h, due to the persistence of the MNBs. This is promising, as proper storage could help not only to improve water quality, but also to reduce costs, as less ozone and therefore less energy may be needed.

Although this work has made a first estimation of the TPs generated in the $O_3$/MNBs processes, more knowledge is needed to better assess the risks associated with the implementation of these large-scale systems, as the TPs generated in hydroxy-radical reactions in some cases may be even more toxic than the starting compounds. Finally, despite the promising results found on the implementation of $O_3$/MNBs systems on a large scale, the impacts of their use at the agronomic level are still largely unknown, from the point of view of plant growth, effects on fertilizers, etc., so these aspects need to be examined further.

**Supplementary Materials:** The following supporting information can be downloaded at: https://www.mdpi.com/article/10.3390/catal13010188/s1, General Information S1: UPLC-QqTOF-MS analysis for water samples; General Information S2: Solid-phase extraction (SPE) protocol; Table S1: Inlet wastewater characterization; Table S2: Suspect list developed under literature research; Table S3: Transformation products detected under experimental conditions.

**Author Contributions:** Conceptualization, methodology, software, validation, formal analysis, investigation, resources and data curation, L.P.-R., A.P.-M. and B.M.-H.; writing—original draft preparation, writing—review and editing, L.P.-R.; visualization, supervision, project administration and funding acquisition, A.J.L.-G., P.A.N.-T., J.J.A.-C. and T.M.-P. All authors have read and agreed to the published version of the manuscript.

**Funding:** The work included in this article was funded by: (i) the strategic project Ris3MUR DIRELMIVID (Total investment: €600,000, EXP: 2120SAE00078), funded by the Consejería de Empresa, Industria y Portavocía, within the framework of the European Regional Development Fund 2014–2020, and (ii) by the strategic project Operational Group Subalma, within the framework of the National Rural Development Program 2014–2020, convened in 2020 by the Spanish Agrarian Guarantee Fund O.A. (FEGA). Total investment €568,758.11 (EAFRD (80%) and by MAPA (20%)). (EXP: O00000226e2000044888).

**Conflicts of Interest:** The authors declare that they have no known competing financial interest or personal relationships that could have appeared to influence the work reported in this paper.

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
