# Peer review of "Full-Scale O3/Micro-Nano Bubbles System Based Advanced Oxidation as Alternative Tertiary Treatment in WWTP Effluents"

_catalysts, doi:10.3390/catal13010188_

Round 1

Reviewer 1 Report

The paper is really interesting and shows a very complete study at pilot scale.

However, need to be carefully polished in the following issues:

-          There are hundreds of English mistakes. A detailed language review is needed.

-          Lines 30-32; Lines 39,42. Have almost the same meaning. Check it.

-          Line 59-60. The toxic effects of microcontaminants is a chronic effect not a short-term effect.

-          There is a lack of several technics and very important details about the pilot plant that are missing and has to be included:

o   The residence time of the water in the ozonisation tank has to be included, as well as the volume of this tank.

o   What is the air quality; oxygen concentration percentage, type of generation?

o   What type is the ozone generator? Corona discharge?

o   What are the injection type and characteristics for MNBs generation?

o   How work the injection? There is a little volume where there is injected and after this flow is mixed with the main one? Add this information.

-           The analysis procedure of Clostridium has to be included.

-          The limit of detection for the MS detector is very high (at least one order of magnitude higher) and similar to other detector types with very lower sensitivity. Why do the authors use it?

-          This reviewer recommends changing By-products to Transformation products. By-products are usually used associated with ozonation to refer to Bromates, NDMA, etc.

Also recommends highlighting the TPs screening and quantification (include in the abstract and in the aim of the introduction section). This part of the work is one of the main strengths of this study.  

-          Line 273-277. Although the theory exposed looks logical (solubility limit), more proofs have to be shown. The ozone solubility in a specific condition can be easily calculated by very well equations. Calculate, compare and after confirm or discard the theory exposed.

-          Line 287-291. If this theory is true, can be easily checked by comparing the different removal kinetics of the contaminants studied due to the reaction with molecular ozone is a highly selective process, i.e., according to the structures their kinetics has to change if the mechanism that predominates is direct or indirect for the different flow conditions. So, the authors have also to show the removal kinetics of different contaminants at the different flow conditions studied and discuss it. In this way, if the theory of the higher turbulence and indirect mechanism at high flows and direct at lower flows can be confirmed or discarded.

-           Figure 3, is not clear what line types are linked with Y axis.

-          Figure 4, the disinfection results are strange. The authors take measurements of ozone concentration in solution and have to include those measurements for all the different conditions tested. With these results, maybe the authors can easily explain the disinfection results obtained and also contaminants removal as well as direct or indirect mechanism predomination at the different flows.

EEO cannot be applied for removal efficiencies lower than an order or magnitude. So, values for the groups 2 and 3 exposed are not true and correct, remove them. Consider how to refocus this aspect, maybe can be applied just to some contaminants. After that, the conclusion that refers to this point has to be also addressed. 

Author Response

All comments can be found in the attached document.

Reviewer 2 Report

The article entitled: “Full scale O3 / Micro-Nano Bubbles system based advanced oxidation as alternative tertiary treatment in WWTP effluents.” Written by Ponce-Robles, L. et al is based on the evaluation of removal efficiency of 12 pharmaceuticals in a WWTP by a O3/MNBs system. The title is clear, the language is correct, and this timely topic is a very interesting and efficiently presented.

Nevertheless, it needs some minor improvements before it can be published in Catalyst. In detail, I have the following comments.

The abstract and introduction are clear, they present the subject and the interest of this work.

The material and methods part is well written and interesting, but the authors haven’t take into account the possibility of adsorption of the transformation compounds on the 0.22 µm PTFE filter used before de LCMS injection, which will reduce the LOD. It would have been an interesting discussion in the text.

The result and discussion part is well argued. However, lines 269-270, when the error is so high, there is no need to give the value with two decimal places.

Conclusion is clear and justified by the previous results.

Publication 68 is actually 2 different publications. Please correct the mistake and check the references in the text.

The name of supplementary files is confusing, S2 appears twice, once as a text and once as a table, please change it.

In SI 1, the mobile phase B is described as 0.1% MeOH. Isn’t it 0.1% FA in MeOH? Please check.

In table S2 the transformation products of Tradazone found in the literature were observed by photolysis. Please check that the degradation compounds are the same after ozonation or find the reference of degradation by O3.

In conclusion, the topic is very interesting and promising, for all the reasons mentioned above, this article should include a minor revision before publication.

Author Response

(The authors gave the same response as above.)

Reviewer 3 Report

This manuscript demonstrates a very interesting work about the applictaion of full scale O3/MNBs system in the treatment of WWTP effluents. It can be accepted after a moderate revision. My comments are listed as follows.

1. It is unnormal to present abstract in two paragraphs. The abstract should be concise and clear in why you did the work, what you found and what it meant.

2.  O3/MNBs system maybe a solution, but the progress in the catalytic ozonation should be mentioned in the Introduction. Here are some advances in these papers: Chemical Engineering Journal, 2021, 404, 127075, Journal of Hazardous Materials, 2022, 431, 128575, Nano Res. 2022, 15, 2961–2970).

3. Is it a commercial system, how much it would cost?

4. ozone bubble range should be measured and highlighted as it is important to capacity and stability. can the bubble size be easily tunned?

5. comparison with other techniques' EEO is suggested to be added and analyzed.

 6. how about the effects of water matrices like NOM, ions, and pH?

Author Response

(The authors gave the same response as above.)

Round 2

Reviewer 3 Report

My concerns have been well addressed. It can be accepted now.

Author Response

The reply to the comments can be found in the attached document.
